# Epidermal Growth Factor Receptor Expression in the Corneal Epithelium

**DOI:** 10.3390/cells10092409

**Published:** 2021-09-13

**Authors:** Joanne L. Peterson, Brian P. Ceresa

**Affiliations:** 1Department of Anatomy, Arkansas College of Osteopathic Medicine, Fort Smith, AR 72916, USA; joanne.peterson@arcomedu.org; 2Department of Pharmacology and Toxicology, University of Louisville, Louisville, KY 40202, USA; 3Department of Ophthalmology and Vision Sciences, University of Louisville, Louisville, KY 40202, USA

**Keywords:** epidermal growth factor receptor (EGFR), cornea, epithelium, wound healing, homeostasis

## Abstract

A properly functioning cornea is critical to clear vision and healthy eyes. As the most anterior portion of the eye, it plays an essential role in refracting light onto the retina and as an anatomical barrier to the environment. Proper vision requires that all layers be properly formed and fully intact. In this article, we discuss the role of the epidermal growth factor receptor (EGFR) in maintaining and restoring the outermost layer of the cornea, the epithelium. It has been known for some time that the addition of epidermal growth factor (EGF) promotes the restoration of the corneal epithelium and patients using EGFR inhibitors as anti-cancer therapies are at increased risk of corneal erosions. However, the use of EGF in the clinic has been limited by downregulation of the receptor. More recent advances in EGFR signaling and trafficking in corneal epithelial cells have provided new insights in how to overcome receptor desensitization. We examine new strategies for overcoming the limitations of high ligand and receptor expression that alter trafficking of the ligand:receptor complex to sustain receptor signaling.

## 1. Cornea Structure and Function

There are two major physiologic roles for the cornea. The first is to refract light onto the retina. Approximately 70% of the refractive power of the eye is from the cornea. In order for this to be done properly, the cornea must remain transparent and pliable. The second role is to serve as a barrier against foreign substances and protect the immune-privileged eye from infectious agents. To mediate these functions, all layers of the cornea must be fully formed.

The cornea is comprised of three cell layers—the epithelium, the stroma, and endothelium. In the human cornea, these layers are separated by the Bowman’s layer [1] and Descemet’s layer [2], respectively (Figure 1A).

### 1.1. Corneal Epithelium

The outermost layer of the cornea is the corneal epithelium. It is approximately 50 μm thick and consists of 4–6 cell layers that undergo constant regeneration and renewal [3]. Over the course of 7–10 days, the entire corneal epithelium is regenerated [4]. Structurally (Figure 1B), the epithelium starts with a basal epithelial layer of columnar cells. As these cells propagate, they differentiate and form polygonal layers of planar cells called wing cells. The outermost two cell layers make up the superficial layer of the corneal epithelium. These cells are flat with an increased surface area to stabilize the tear film. To maintain a proper level of cell hydration, cell–cell communication, and selective permeability, adhesion of the cells is facilitated by tight junctions amongst the superficial cells, desmosomes between neighboring wing cells and between wing and basal cells, gap junctions and desmosomes between basal cells, and hemidesmosomes between the basal cells and Bowman’s layer [5,6,7].

### 1.2. Stroma

The stroma makes up most of the cornea’s thickness (~450–500 μm) and is an organized network of collagen fibrils and proteoglycan aggregates that contain specialized fibroblasts called keratocytes [8,9]. Keratocytes secrete collagen, glycosaminoglycans, and matrix metalloproteinases [10]. While there are a variety of collagens that comprise the stroma, types I, III, IV, and V are the most predominant [11]. Together, the alignment of small and large proteoglycans provide the clarity and hydration that are necessary to keep the stroma clear and pliable.

Both the epithelium and stroma are highly innervated, with a nerve density that is 300–600 times greater than the skin [12]. Corneal nerves are derived from the ophthalmic division of the trigeminal nerve. Fibers at the corneoscleral limbus travel through the stroma and penetrate through the Bowman’s layer to enter the epithelium [13,14,15] (Figure 1B). They mediate the sensations of touch, pain, and temperature, and control blinking and the tear reflex. Both responses help protect the epithelial layer from foreign insults. Corneal nerves play a role in wound healing as well. Corneal neuropathy, such as from chronic dry eye disease, diabetes, or trigeminal neuralgia, decreases the normal responses to corneal epithelial perturbations (i.e., blinking and tearing), leading to the development of persistent corneal erosions and potential corneal blindness [16].

### 1.3. Endothelium

The endothelial layer is critical for proper hydration of the stromal layer [17]. This single layer of cells uses ionic pumps to regulate the movement of nutrients and water from the aqueous humor to the cornea [18]. Excessive hydration of the stroma can disrupt the structure of the stroma and result in scattering of light and distorted vision [19]. Endothelial cells do not divide, so if cells are lost due to disease (i.e., Fuchs’ dystrophy) or injury, the remaining cells undergo morphological changes that allow them to fill in the acellular areas and fulfill the role of the missing cells [20].

### 1.4. Bowman and Descemet’s Layers

The two membranes, Bowman’s [1] and Descemet’s [2], are acellular interfaces between the cellular layers. The Bowman’s layer is approximately 8–12 μm thick [21] and is comprised mainly of collagen type I, but also contains collagens type III, V, and XII. It is found in humans [22] and most non-human primates [23], but notably is not found in mice [24]. At this point, the role of the Bowman’s layer is not clear [1]. The fact that it can be removed in photorefractive keratectomy and is not found in all species has complicated our understanding of its role.

The Descemet’s layer is 3–5 μm thick and composed of extracellular components common to most basement membranes: laminins, collagen type IV, nidogen-1 and nidogen-2, and perlecan [2]. These components can bind transforming growth factor β1 and β2 (TGFβ1 and TGFβ2) [25] and platelet-derived growth factor (PDGF) receptor agonists (PDGF-AA and PDGF-BB) [26] to prevent movement of these growth factors, and possibly others, from the aqueous humor to the stroma, and, ultimately, fibrosis of the stroma [2].

## 2. Corneal Epithelial Perturbations

Based on its structural proximity to the environment, virtually all insults to the eye affect the corneal epithelium. These include trauma, infections, and diseases. The high degree of innervation can make traumatic wounds painful. Further, by exposing the interior chamber of the eye to the environment, there is an increased risk for infection that, if not properly treated, could lead to blindness.

Delays in restoring the corneal epithelium could have significant consequences. In addition to excluding viruses, bacteria, and small particles from the eye, an intact epithelial layer prevents entry of transforming growth factor β1 and β2 (TGFβ1 and TGFβ2) from the stroma. TGFβ binds to its cognate receptor on keratocytes and promotes their differentiation to myofibroblasts [27]. These fibroblasts form scar tissue and obscure vision.

Despite these known risks and the frequency with which corneal trauma occurs, there are no FDA-approved therapies to promote wound healing, largely due to our incomplete understanding of how to enhance regeneration of the damaged corneal epithelium.

### 2.1. Trauma

Trauma to the cornea is a relatively common occurrence and can range from complex, penetrating wounds, to more superficial scratches of the epithelial layer. Some of the corneal insults can be deliberate, such as surgical procedures including cataract surgery (replacement of the lens that becomes cloudy with age) or keratectomy (a corrective procedure to reshape the cornea to improve vision) that involve full penetration of the cornea [28]. Others are the result of chemical, thermal, or physical trauma to the eye and mainly impact the outermost layers [29,30]. Still others can be a consequence of disease (diabetes) [31,32,33,34] or drug use (anti-cancer drugs) [35,36,37] in which patients contend with an increase in corneal erosions. The extent of these injuries can vary based on the duration and exposure of the external agent. Superficial wounds may have minimal complications and quickly heal, whereas injuries that penetrate to the stroma will not only take longer to heal but will also run the risk of developing into a persistent injury, becoming infected, or leading to corneal scarring.

### 2.2. Infections

The corneal epithelium can be infected by viruses (i.e., herpes simplex virus-1 [38], herpes zoster [39]), bacteria (i.e., *Staphylococcus aureus* and *Pseudomonas aeruginosa* [40]), and fungi (i.e., *Fusarium*, *Aspergillus*, and *Candida* [41]). These infections can result in ocular pain and redness, reduced vision, tearing and discharge from the eye, and light sensitivity (photophobia) [42]. Typically, infections can be resolved with the appropriate chemotherapeutic therapy alongside treating the secondary symptoms. The prognosis for most infections is good with a prompt diagnosis and quick treatment [43].

### 2.3. Corneal Diseases

In addition, there are a number of non-infectious and non-hereditary corneal epithelial diseases as well [44]. Although not comprehensive, this list includes Thygeson’s superficial punctate keratitis, neurotrophic keratitis, filamentary keratitis, factitious keratoconjunctivitis, vortex keratopathy, corneal epithelial deposits (adrenochrome and fluoroquinolone deposition), and corneal epithelial edema. Each pathology is distinguished by its presentation and has a specific pharmacological intervention that helps resolve the condition. Frequent complications of this group of diseases includes corneal sensitivity, photophobia, dry eye disease, deposition of degenerated epithelial cells, or structural perturbations in the homogeneity of the corneal epithelium.

## 3. Regulation of the Corneal Epithelial Homeostasis

There are a number of growth factor receptors and cytokine receptors whose signaling contribute to the restoration and homeostasis of the corneal epithelium [45]. This list includes receptors for platelet-derived growth factor [46], hepatocyte growth factor (HGF) [47], colony-stimulating factor 1 [48], keratinocyte growth factor [49], fibroblast growth factor [50,51], transforming growth factor-β2 [52], nerve growth factor [53], interleukin-1 [54], and interleukin-6 [55]. However, the epidermal growth factor receptor (EGFR) stands out because in rodent models, stimulation with epidermal growth factor (EGF) can accelerate corneal epithelial wound healing up to two-fold [56,57] and the addition of EGFR inhibitors (i.e., AG1478) to wounded corneal epithelium in mice prevents wound healing [58].

These experimental data are supported by reports from the clinic that patients taking EGFR inhibitors (i.e., cetuximab, erlotinib, gefitinib) for the treatment of certain cancers have an increased incidence of corneal perturbations, including punctate keratopathy, dry eye, blepharitis, conjunctivitis, and trichiasis [35,36,59,60,61]. These are typically limited to the ocular surface and are readily treatable with topical lubricants and antibiotics. The chief concerns are increased discomfort for an already ill patient and in the most severe cases, the likelihood for infection.

### 3.1. EGFR Expression in Corneal Epithelial Cells

The fact that corneal epithelial homeostasis is dependent on the EGFR is not entirely surprising. The relative distribution of the EGFR in corneal epithelial cells has been reported previously [57]. Indirect immunofluorescent staining of the EGFR shows the receptor is more concentrated along the plasma membrane in the basal epithelial layers, with a decrease in the intensity of receptor staining as the cells differentiate. In addition, following wounding, the receptor moves from the plasma membrane to cytosolic locations, consistent with ligand-mediated receptor internalization [62].

Immortalized human corneal epithelial (hTCEpi) cells have ~1,100,000 ^125^I-EGF binding sites/cell and binding with negative cooperativity, as seen with the EGFR in other cell lines [63] (Figure 2A). Interestingly, the apparent Kd of ^125^I-EGF binding is 1.52 nM, with only modest differences between the high- and low-affinity states (1.06 nM and 2.85 nM, respectively). While the Kd for low-affinity binding was in line with reports in other cell lines, the Kd for high-affinity binding sites differs by about 10-fold [63,64,65]. Approximately ~20% of the receptor binding sites in the corneal epithelial cells are high affinity, which is higher than other cell lines (3–14% range) [63,64,66]. While the basis for these disparities is not clear, these differences may reflect cellular changes in receptor methylation [67], the presence of corneal epithelial specific effectors, or perturbations in receptor trafficking [68]. Comprehensive Scatchard analysis has not been performed in primary corneal epithelial cells due to limitations in the availability and the heterogeneity of human tissue. However, EGFR protein and mRNA are comparable in immortalized and primary cells [56,69].

### 3.2. Ocular Expression of EGFR Ligands

It is worth noting that no other healthy tissue in the human body has been reported to express EGFRs at this high of a density. However, some cancer-derived cell lines express EGFRs at this density (Table 1). A representative sample of non-cancerous cell lines indicates EGFR densities are in the range of 50,000 to 70,000 receptors/cell. Importantly, those cells have many of the same EGFR-mediated responses seen in corneal epithelial cells and are associated with tissue restoration and homeostasis.

Complementing the robust level of receptor expression are high basal levels of EGF in tear fluid. Tear fluid consists of a mixture of aqueous solutions and lipids and provides a film over the cornea to maintain hydration and lubrication of the ocular surface. In addition, it flushes away contaminating particles, protects against pathogens, nourishes the cells, and provides necessary electrolytes and metabolites [84]. Tears are produced in the lacrimal cells and the components of the tears are produced by the lacrimal gland as well as the corneal epithelial cells [57,85,86]. There is a constant basal production of tears that can increase in response to stimuli [87].

Several studies of tear fluid from healthy human volunteers (i.e., no known ocular pathologies) show that EGF levels are approximately 2 ng/mL (ranging from 0.7 to 8.4 ng/mL) under normal conditions [56,88,89,90]. It is important to note that this concentration of EGF is close to the reported Kd of the ligand for the receptor (0.32 nM) [87]. Thus, under steady-state conditions, there are sufficient levels of EGF to occupy approximately half of the EGFRs. In healthy human volunteers, the other EGFR ligands, namely transforming growth factor-α (TGFα), heparin-binding epidermal growth factor (HB-EGF), betacellulin (BTC), epiregulin (EPR), and amphiregulin (AR), were much less prevalent. These other ligands were either below the limit of detection or had average concentrations that were more than 20-fold less than their Kd for the EGFR [56].

Increases in other EGFR ligands have been documented using in vitro, ex vivo, and in vivo models of corneal epithelial wounding. For instance, cultured corneal epithelial cells subjected to mechanical wounding have an increase in EGFR-dependent cell migration that is ablated by the addition of anti-HB-EGF antibodies [91]. The addition of the matrix metalloprotease inhibitor, GM 6001, prevents activation of the EGFR, consistent with a block in processing (not production) of the HB-EGF that drives receptor activity [91]. Similar results were observed using ex vivo porcine corneas [86]. Other studies demonstrate that wounded cornea epithelial cells produce high levels of adenosine triphosphate (ATP) (~0.5 μM), which in turn activates its cognate purinergic receptor [92]. This signals the activation of Phospholipase Cβ (PLCβ), which stimulates the appropriate matrix metalloproteinase that converts the pro-HB-EGF to the soluble form [92,93].

In addition, the mRNA of other EGFR ligands is upregulated by trauma to the mouse corneal epithelium. AR, HB-EGF, and TGFα increase following debridement of the corneal epithelium [57]. In addition, EGF mRNA increases ~50% in the lacrimal gland following wounding [85]. In vivo studies indicate that the mRNA levels of hepatocyte growth factor and keratocyte growth factors increase as well, following wounding [85,94]. Further investigation will likely reveal increases in additional growth factors.

## 4. Physiologic Role of the EGFR and ErbB Family in Corneal Epithelial Homeostasis

### 4.1. EGFR in the Corneal Epithelium

Given that EGFR activity is necessary and sufficient to drive wound healing and tissue homeostasis, it seems that exogenous EGF would be an excellent therapy for treating corneal wounds. Unfortunately, this is not always the case. The success of EGF treatment is dependent on the cause of the wound. For example, patients with corneal abrasions and epithelial lesions saw marked healing with EGF treatment [95], as did corneal erosions associated with the use of cetuximab (an EGFR inhibitor) to treat cancer [59,60,96] and traumatic corneal ulcers [96]. In other studies, EGF did not promote improvement. Topical EGF does not help patients with herpes simplex dendritic ulcers, bullous keratopathies, stromal keratitis, or penetrating keratoplasty [95,97,98].

Although the exact molecular basis for the discrepancy between the laboratory and the clinic is not fully understood, several hypotheses have been put forth. The first suggests that the laboratory models of corneal wound healing are more sensitive. While some of the nuanced details of the wound healing protocols may change (i.e., wound size, method for epithelial debridement, etc.) a common feature is the use of anesthetics prior to wounding [99]. Studies have shown that the frequently used anesthetic agent, xylazine (an α_2_ adrenergic agonist), can decrease tear production [100]. This likely reduces the distribution of the endogenous EGF to the cornea surface and the drop in basal ligand distribution increases the response to exogenous EGF.

Alternatively, since in humans EGF concentrations are close to the Kd of the ligand for the receptor, the EGFR response may be limited due to high receptor occupancy and more EGF does not increase the response [56]. In addition, the high levels of EGF in humans may lead to desensitization of the EGFR, making more EGF counterproductive [62,69]. If mice have lower EGF levels in tear fluid, the EGFRs may be primed to be more responsive. Due to the intrinsically low tear volume, murine EGF concentrations have not been determined.

Another possible explanation is there are pharmacokinetic differences in growth factor delivery to the corneal epithelium due to proteases or binding proteins. This would limit the amount of ligand that achieves EGFR binding.

### 4.2. ErbB Family Members in the Corneal Epithelium

Numerous lines of evidence point to EGF and EGFR being central regulators of corneal epithelial homeostasis. However, the role of closely related ligand and receptor family members cannot be overlooked. ErbB2, ErbB3, and ErbB4 are also members of the ErbB receptor tyrosine kinase family. They are similar in size and structure, but differ in tissue distribution, enzymatic activity, and activating ligands [101].

Both ErbB2 and ErbB3 are expressed in the corneal epithelium with a similar receptor distribution as that reported for EGFR [102]. Although there were some early reports that ErbB4 was expressed in the cornea epithelium [103], a more detailed analysis of mRNA levels or antibody specificity indicate that ErbB4 is not expressed [56] (Figure 3A).

Experiments using cultured, immortalized corneal epithelial cells have demonstrated that ErbB2 and ErbB3 activity contribute to corneal epithelial wound healing and homeostasis. Mechanical perturbation of corneal epithelial cells induces ErbB2 phosphorylation and knockdown of ErbB2 reduces cell migration in scratch and Boyden chamber cell migration assays [104]. Like the EGFR-specific inhibitors, ErbB2 inhibitors (i.e., trastuzumab) have also been associated with ulcerative keratitis [105,106].

The effects of ErbB3 are not as well described. Since ErbB3 is kinase impaired [107], the development of ErbB3 inhibitors has lagged behind those for EGFR and ErbB2. However, ErbB3-specific antibody inhibitors do exist [108]. Using these inhibitors and genetic approaches, it has been shown that ErbB3 signaling can promote corneal epithelial cell migration [109].

### 4.3. Specific Effects of EGFR Ligands on the Corneal Epithelial Cells

Despite the fact that EGF is the only EGFR ligand detected in the unstimulated tear fluid of healthy volunteers, other naturally occurring ligands are logical candidates for stimulating EGFR-mediated corneal epithelial homeostasis. An in vitro analysis of the other EGFR ligands demonstrates HB-EGF, BTC, and TGF-α can accelerate wound healing as well, if not better, than EGF [56] (Figure 3B).

BTC seemed particularly promising, as in vitro studies indicated BTC was ~20% better than EGF at promoting wound closure [56]. In addition, BTC-treated corneal epithelial cells were more migratory than those treated with EGF, despite lower levels of receptor phosphorylation [109]. The molecular basis for this difference was not due to ligand affinity for the receptor (BTC = 0.5 nM and EGF = 0.3 nM) [110,111], but the dimerization partner. EGF binding promotes the homodimerization of EGFRs. In corneal epithelial cells, BTC binds the EGFR, which biases the receptor to heterodimerize with ErbB3 [109]. In EGFR:ErbB3 heterodimers, the attenuated ErbB3 kinase activity reduces the amount of phosphorylated EGFR, but increases the amount of phosphorylated ErbB3. It is ErbB3’s phosphotyrosines that are thought to recruit and activate the phosphatidyl inositol 3-kinase needed for cell migration.

In addition, TGF-α was thought to be a strong therapeutic candidate. It has been well documented that TGF-α stimulation of the EGFR promotes internalization and recycling of the ligand:receptor complex [112,113]. Instead of promoting receptor downregulation (as is the case following EGF treatment), the TGF-α:EGFR complex recycles back to the plasma where it can be re-activated. Bypassing the degradative pathway makes TGF-α a more efficacious activator of the EGFR. This holds true in immortalized and primary corneal epithelial cells [62].

Despite the predicted enhanced responses from BTC and TGF-α in cell biology assays, this did not translate to in vivo wound healing using murine models (Figure 3B). Although murine models are regarded as excellent models for corneal wound healing [28], we cannot discount the major anatomical difference between the mouse and human cornea: mice lack a Bowman’s layer. In most studies, this has not been an issue. However, it is possible that these ligands may elicit a biological response that favors epithelial cell binding to the stroma over the Bowman’s layer. Further, we do not know if BTC and TGF-α are stable in the mouse tear fluid. Are there other components of the tear fluid that impact function in these animals? Does the mouse have differing levels of ErbB3 expression that might affect BTC function [109]? Does TGF-α promote EGFR recycling in the mouse cornea epithelial cells the way it does in human corneal epithelial cells? Does the production of tears in the mouse impact delivery of the ligands? In summary, work needs to be carried out to establish whether the differences between in vitro and in vivo analysis are a technical or a biological issue.

### 4.4. Negative Consequences of Sustained EGFR Activity

Although EGFR activity promotes corneal epithelial cell migration and proliferation, it is important to note that increasing the magnitude and duration of EGFR may not elicit only positive effects. Rodent corneal wound healing experiments indicate that daily application of EGF over the course of 35 days can be deleterious [114]. Topical EGF leads to a three-fold increase in collagen deposition immediately after wounding and EGF administration. However, 21 days post-wounding, the EGF-treated eyes had less collagen than the phosphate-buffered saline control treatment. The bimodal response to EGF treatment is possibly due to downregulation of EGFR signaling due to receptor desensitization. The levels of protease and collagenase activity were unchanged at time points (14–21 days) with reduced wound collagen.

Administration of recombinant EGF as a sustained release pellet in the cornea can lead to neovascularization [115]. In addition, studies have shown that high levels of tear EGF are associated with hypertrophy of the meibomian gland duct and cause meibomian gland hyperplasia [116]. Together, these data highlight the need for caution when treating corneal wounds with exogenous EGF.

## 5. Pharmacological Approaches to Promote Corneal Epithelial Homeostasis

Since enhanced ligand stimulation does not reliably restore corneal epithelial homeostasis, other strategies are being developed. Rather than trying to increase the EGFR-mediated response, an alternative approach is to prevent desensitization of the activated EGFR (Figure 3C). Like many cell surface receptors, the EGFR internalizes following ligand binding. When EGF binds the receptor, the ligand:receptor complex enters the cell via clathrin-coated pits that pinch off from the membrane to give rise to clathrin-coated vesicles. Once inside the cell, the clathrin dissociates and the resulting intermediate vesicle fuses with the early endosome. The ligand and receptor are trafficked through the late endosome to the lysosome for degradation [117]. Thus, an alternative approach is to divert the ligand:receptor complex from the lysosome to keep the receptor active for longer. Thus, more EGFR signaling is achieved by increasing the duration of receptor signaling, rather than the magnitude.

The best strategies for disrupting EGFR endocytic trafficking are ones that do not affect the constitutive trafficking of the receptor. Disrupting constitutive receptor trafficking can lead to the accumulation of receptors within the endocytic pathway and reduce the cell surface levels of the receptor available for stimulation. Ligand-independent intracellular accumulation of the EGFR has been observed following the inhibition of RAB5, RAB7, or TSG101 function [118,119].

The E3 ubiquitin ligase, c-Cbl, has emerged as a viable target for prolonging EGFR signaling. c-Cbl ubiquitylates the EGFR in response to ligand treatment and regulates the rate of EGF:EGFR complex internalization and degradation [120]. Further, the loss of c-Cbl enhances EGFR-dependent cell migration and in vivo wound healing [69]. Knockdown of c-Cbl does not affect the steady-state distribution of the EGFR, and does not impinge upon its potential to signal [119]. Together, these data indicate that antagonizing c-Cbl is a viable approach for enhancing EGFR signaling that bypasses the limitation of receptor desensitization.

## 6. Conclusions

EGFR activity is a driver of corneal epithelial homeostasis. Stimulation of the receptor promotes wound healing and inhibition of the receptor prevents it. Due to intrinsic regulatory mechanisms that prevent sustained EGFR signaling, EGFR agonists have not been a reliable therapeutic strategy for resolving perturbations of the corneal epithelium. However, as we gain a more complete understanding of the molecular mechanisms that regulate EGFR signaling, new approaches are being developed that will overcome the intrinsic regulatory mechanisms.

## Figures and Tables

**Figure 1 cells-10-02409-f001:**
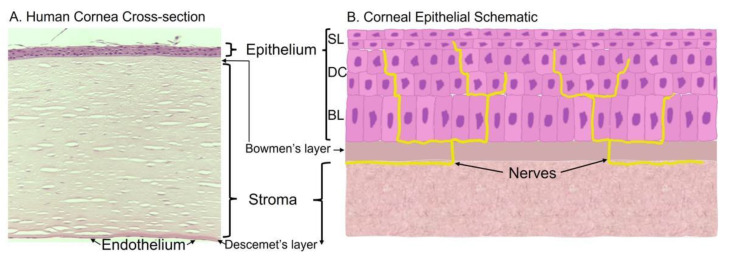
Histology and schematic of the cornea. (**A**) Human cornea obtained from cadaver donation. Corneal tissue was placed in 4% paraformaldehyde overnight, washed, and embedded in paraffin for histological sectioning and hematoxylin and eosin staining to visualize the various cell layers and membranes. (**B**) Schematic of the corneal epithelium that highlights the epithelial layers and innervation; BL = basal epithelial cells, DC = differentiated cells, SL = superficial layer.

**Figure 2 cells-10-02409-f002:**
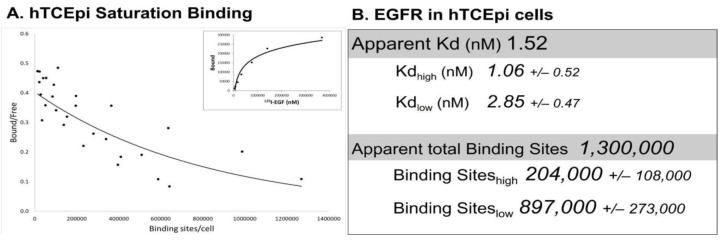
Saturation binding in hTCEpi cells. (**A**) Saturation binding assays were performed using adherent immortalized corneal epithelial (hTCEpi) cells [70] using ^125^I-EGF (Perkin Elmer, Waltham, MA, USA), as previously described [71,72]. Shown are cummulative data from 4 independent experiments (6–7 data points/experiment) that were subjected to Scatchard transformation. The concave-up binding curve (solid line) is consistent with two affinity sites. Saturation binding from one representative experiment is shown in the inset. (**B**) The apparent Kd and total binding were generated from the summative data of all four experiments. High- and low-affinity binding sites were calculated from each experiment with PRISM utilizing a non-linear, two-site binding analysis. Data are presented as the average ± S.E.M. (*n* = 4).

**Figure 3 cells-10-02409-f003:**
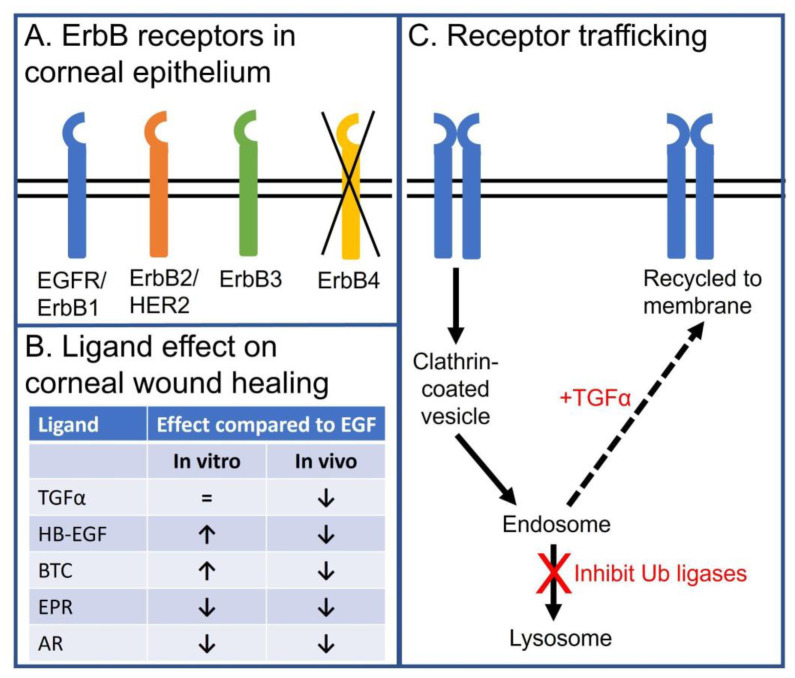
Schema summarizing the role of EGFR and ErbB family members in the corneal epithelium. (**A**) Schematic depicting expression of EGFR (ErbB1), ErbB2 (Her2), and ErbB3, but not ErbB4, in corneal epithelium. (**B**) Table summarizing the effects of endogenous EGFR ligand on in vitro and in vivo corneal epithelial wound healing. (**C**) Schematic depicting how EGFR endocytic trafficking affects receptor signaling in corneal epithelial cells. EGF stimulation promotes internalization of the EGF:EGFR complex to the early endosome. During internalization, the receptor is ubiquitylated and targeted for trafficking to lysosome for degradation. EGFR degradation can be diverted from a degradation pathway by promoting receptor recycling. This can be achieved by stimulating with TGFα, a ligand that dissociates in the mildly acidic environment of the early endosome and has reduced receptor ubiquitylation, or inhibiting the ubiquitin ligases of the EGFR in corneal epithelial cells (i.e., c-Cbl and Cbl-b).

**Table 1 cells-10-02409-t001:** EGFR density in selected cell lines.

Cells	EGFRs/Cell	Reference
**Non-cancerous cell lines**		
hTCEpi (human)	1,300,000	Figure 2
Oral mucosa	200,000	[73]
Skin fibroblasts	51,000–70,000	[71,74,75]
Primary corneal endothelium	40,000	[76]
Blood cells	7800–25,400	[77]
Gastric smooth muscle	24,000	[78]
**Cancer cell lines**		
A431 cell (epidermoid carcinoma)	1,500,000–2,600,000	[74,75,79]
MDA-MB-468 cells (mammary gland)	1,900,000	[74]
Moser-1 (colon cancer)	295,700	[80]
HepG2 (liver carcinoma)	180,000	[81]
HT29 (colon cancer)	120,000	[80]
HeLa (cervical adenocarcinoma)	43,500	[82]
MCF-7 (mammary epithelial)	2800–10,000	[83]
KM12SM (colon cancer)	7000	[80]

## Data Availability

The data presented in this study are available on request from the corresponding author.

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
