# Peer review of "Epidermal Growth Factor Receptor Expression in the Corneal Epithelium"

_cells, 2021, doi:10.3390/cells10092409_

Round 1

Reviewer 1 Report

1.1 Typically the corneal epithelium is thought of as being pseudostratified with the greatest abaundance of tight junctions in the apical layer and not the basal layer. This was elegantly shown by an early study of Dr. Zieske. Please rectify this. Desmosomes, adherens junctions and gap junctions are more prevalent in the lower layers, with hemidesmosomes adhering the basal cells to the basement membrane.

1.3 Keratocytes secrete collagens and not just one collagen type. The stroma is an excellent example of the different collagens.

To say that corneal nerves originate in the stroma is misleading. Please check corneal nerve reviews.

1.4 The endothelium also regulates other growth factors and these have different avidities to matrix molecules in the stroma.

3. Acceleration of wound healing is not "all good" as it can be disregulated as shown in a number of studies. Both sides should be reflected here.

3.1 There is a discussion of primary corneal epithelial cells but the data presented in the Table is of a cell line. Please discuss.

3.2 Early studies have shown that the bursting strength and concentration of EGF is bimodal demonstrating a common phenomena of growth factor binding. Please discuss.

4.2 The authors do not discuss at all the importance of specific residues of the EGFR (Erb1). This would enhance the study and demonstrates that some residues are systemically on while others when deleted alter wound healing and cell migration.

Author Response

We thank the reviewers for their considerate and constructive comments on our article. We feel their input has resulted in a much clearer manuscript that reflects our oveall goal. Below are the verbatim comments from each reviewer with our response below.

  1. 1.1 Typically the corneal epithelium is thought of as being pseudostratified with the greatest abundance of tight junctions in the apical layer and not the basal layer. This was elegantly shown by an early study of Dr. Zieske. Please rectify this. Desmosomes, adherens junctions and gap junctions are more prevalent in the lower layers, with hemidesmosomes adhering the basal cells to the basement membrane.

We have modified the text and included the appropriate reference. (lines 39-41, 42-47)

  1. 1.3 Keratocytes secrete collagens and not just one collagen type. The stroma is an excellent example of the different collagens.

We have modified the text and include references to the specific collagens. (lines 52-53)

  1. To say that corneal nerves originate in the stroma is misleading. Please check corneal nerve reviews.

We have changed the wording in the section to make it more accurate. (lines 56-58)

  1. 1.4 The endothelium also regulates other growth factors and these have different avidities to matrix molecules in the stroma.

The edited manuscript includes discussion or additional growth factors. (lines 88-93)

  1. 3. Acceleration of wound healing is not "all good" as it can be disregulated as shown in a number of studies. Both sides should be reflected here.

We thank the reviewer for highlight this point. We have included an additional section (4.4) that addresses some of the negative consequences associated with increased EGFR signaling. (lines 324-338)

  1. 3.1 There is a discussion of primary corneal epithelial cells but the data presented in the Table is of a cell line. Please discuss.

We have edited the text such that it is clear the binding studies were performed in an immortalized cell line. We note that similar levels of EGFR protein and mRNA were observed in primary corneal epithelial cell lines as well. (line 167)

  1. 3.2 Early studies have shown that the bursting strength and concentration of EGF is bimodal demonstrating a common phenomena of growth factor binding. Please discuss.

We have included a new section to the manuscript that highlights how EGF treatment causes an increase in wound collagen levels immediately after EGF treatment that attenuates with time. (lines 327-333) In addition, we discuss the high and low affinity states of the EGFR. (lines 167-179)

  1. 4.2 The authors do not discuss at all the importance of specific residues of the EGFR (Erb1). This would enhance the study and demonstrates that some residues are systemically on while others when deleted alter wound healing and cell migration.

We appreciate the reviewer’s comment. There are very few studies of functionally important domains of the EGFR in corneal wound healing. This is likely due to the effort that is involved in removing endogenous EGFRs and replacing them with mutant receptors to assess functional importance. However, there is one example of the importance of EGFR residues in the cornea - the so-called Velvet mouse strain (https://www.jax.org/strain/006926). This mouse was generated through by chemical mutagenesis (i.e. a randomly induced mutation) that happened to be the EGFR. It has a D833G mutation in the ATP binding domain and most kinase activity is abolished. The mouse does experience issues with corneal opacity, but in short, it is unclear if this is truly due to reduced EGFR signaling and likely does not impact the corneal epithelium directly. We feel it is best to not include these types of studies in this review to avoid mis-leading the reader.

Reviewer 2 Report

The manuscript of Peterson & Ceresa is a comprehensive review highlighting the role of EGFR  in corneal epithelium. The text is clear and unambiguous. Overall, the literature is well referenced and relevant.

However, I have some suggestions to improve the manuscript.

1.  While the title suggests that the focus is on EGFR expression, the review almost covers other relevant issues. So, the title should be adequate to the review content. 

2. The abstract should also specify the most recent evidence implicating EGFR  in cornea homeostasis and selected corneal disturbers (cited throughout the review).

3. Figure 1 is informative. However, the two membranes, i.e., Bowman’s and Descemet’s, should be equally labeled and described (same letter size and color).

4.  My most important criticism:
The content of item 3.1, entitled EGFR expression in corneal epithelial cells, should be revised. Are the results of figure 2 unpublished data from previous reports (refs 59, 60, 61)? To avoid the loss of focus, the authors should maintain a writing style expected for review articles instead of experimental articles  (as presented, the text includes some methodological details, a description of results, and a brief discussion). 

5. To better highlight the functions of the epidermal growth factor receptor (EGFR/ErbB) pathway in cornea and corneal disturb, the authors should design a schema summarizing the evidence of the EGFR ligands on the corneal epithelial cells, as well as the effect of its inhibitors. This would help to highlight some of the observations described in items 4 and 5. 

Author Response

We thank the reviewers for their considerate and constructive comments on our article. We feel their input has resulted in a much clearer manuscript that reflects our oveall goal. Below are the verbatim comments from each reviewer with our response below.

Reviewer #2

  1.  While the title suggests that the focus is on EGFR expression, the review almost covers other relevant issues. So, the title should be adequate to the review content.

We appreciate this suggestion and the acknowledgement of our efforts to provide context in our article. However, in keeping with the theme of the special issue of the journal (‘EGF Receptor Trafficking Pathways: From Epithelial Cell Functions to Epithelial Carcinomas’), we feel our title accurately captures the main point of the review.

  1. The abstract should also specify the most recent evidence implicating EGFR  in cornea homeostasis and selected corneal disturbers (cited throughout the review).

We have edited the abstract to better reflect the content of the article. We appreciate the suggestion.

  1. Figure 1 is informative. However, the two membranes, i.e., Bowman’s and Descemet’s, should be equally labeled and described (same letter size and color).

We have modified the figure for internal consistency.

  1.  The content of item 3.1, entitled EGFR expression in corneal epithelial cells, should be revised. Are the results of figure 2 unpublished data from previous reports (refs 59, 60, 61)? To avoid the loss of focus, the authors should maintain a writing style expected for review articles instead of experimental articles  (as presented, the text includes some methodological details, a description of results, and a brief discussion).

We appreciate the point of the reviewer. They are, in fact, correct that Figure 2 represents data that has not been previously published. We felt this was warranted based on the important information it brings to the field of EGFR biology. This is an exceptional level of EGFR expression for a non-cancerous tissue.

We had edited the text to maintain a writing style that is consistent with the rest of the article and only included experimental details, results and discussion when absolutely required. (lines 167-179)

  1. To better highlight the functions of the epidermal growth factor receptor (EGFR/ErbB) pathway in cornea and corneal disturb, the authors should design a schema summarizing the evidence of the EGFR ligands on the corneal epithelial cells, as well as the effect of its inhibitors. This would help to highlight some of the observations described in items 4 and 5. 

We appreciate this suggestion. We have included a new Figure 3 in the revised manuscript that summarizes several important points about EGFR signaling in the corneal epithelium.

Round 2

Reviewer 1 Report

The author has unfortunately not put the review through spell check and there are numerous spelling mistakes. This is not tolerable and highly irritating to the reviewer who must document them. Please check the  words and or sentences in the following lines.

line 44 - 2 words, line 52 still do not mention type V collagen and the stroma is not an orchestra, line 56 2 words including ophthalmic!, line 85-don't end with the word is, line 88 - rewrite, line 89 - it is laminin and not lamin!, line 334-5-rewrite.
